# Exploratory Pilot Study of Circulating Biomarkers in Metastatic Renal Cell Carcinoma

**DOI:** 10.3390/cancers12092620

**Published:** 2020-09-14

**Authors:** Ilaria Grazia Zizzari, Chiara Napoletano, Alessandra Di Filippo, Andrea Botticelli, Alain Gelibter, Fabio Calabrò, Ernesto Rossi, Giovanni Schinzari, Federica Urbano, Giulia Pomati, Simone Scagnoli, Aurelia Rughetti, Salvatore Caponnetto, Paolo Marchetti, Marianna Nuti

**Affiliations:** 1Laboratory of Tumor Immunology and Cell Therapy, Department of Experimental Medicine, Policlinico Umberto I, “Sapienza” University of Rome, 00161 Rome, Italy; ilaria.zizzari@uniroma1.it (I.G.Z.); alessandra.difilippo@uniroma1.it (A.D.F.); aurelia.rughetti@uniroma1.it (A.R.); marianna.nuti@uniroma1.it (M.N.); 2Division of Oncology, Department of Radiological, Oncological and Pathological Science, Policlinico Umberto I, “Sapienza” University of Rome, 00161 Rome, Italy; andrea.botticelli@uniroma1.it (A.B.); alain.gelibter@uniroma1.it (A.G.); federica.urbano@uniroma1.it (F.U.); giuliapomati@tiscali.it (G.P.); simone.scagnoli@uniroma1.it (S.S.); salvo.caponnetto@uniroma1.it (S.C.); paolo.marchetti@uniroma1.it (P.M.); 3Division of Medical Oncology B, San Camillo Forlanini Hospital Rome, 00149 Rome, Italy; fabiocalabro1@alice.it; 4Department of Medical Oncology, Fondazione Policlinico A.Gemelli Rome, 00168 Rome, Italy; Ernesto.rossi@guest.policlinicogemelli.it (E.R.); giovanni.schinzari@policlinicogemelli.it (G.S.); 5Division of Oncology, Department of Clinical and Molecular Medicine, Ospedale Sant’Andrea, “Sapienza” University of Rome, 00189 Rome, Italy

**Keywords:** TKIs, mRCC, biomarkers, soluble factors

## Abstract

**Simple Summary:**

The identification of biomarkers in response to therapeutic treatment is one of the main objectives of personalized oncology. Predictive biomarkers are particularly relevant for oncologists challenged by the busy scenario of possible therapeutic options in mRCC patients, including immunotherapy and TKIs. In fact the activation of the immune system can determine the outcome and success of the different therapeutic strategies. In this study we evaluated changes in the immune system of TKI mRCC-treated patients defining immunological profiles related to response characterized by specific biomarkers. The validation of the proposed immune portrait to an extended number of patients could allow characterization and selection of responsive and non-responsive patients from the beginning of the therapeutic process.

**Abstract:**

With the introduction of immune checkpoint inhibitors (ICIs) and next-generation vascular endothelial growth factor receptor–tyrosine kinase inhibitors (VEGFR–TKIs), the survival of patients with advanced renal cell carcinoma (RCC) has improved remarkably. However, not all patients have benefited from treatments, and to date, there are still no validated biomarkers that can be included in the therapeutic algorithm. Thus, the identification of predictive biomarkers is necessary to increase the number of responsive patients and to understand the underlying immunity. The clinical outcome of RCC patients is, in fact, associated with immune response. In this exploratory pilot study, we assessed the immune effect of TKI therapy in order to evaluate the immune status of metastatic renal cell carcinoma (mRCC) patients so that we could define a combination of immunological biomarkers relevant to improving patient outcomes. We profiled the circulating levels in 20 mRCC patients of exhausted/activated/regulatory T cell subsets through flow cytometry and of 14 immune checkpoint-related proteins and 20 inflammation cytokines/chemokines using multiplex Luminex assay, both at baseline and during TKI therapy. We identified the CD3^+^CD8^+^CD137^+^ and CD3^+^CD137^+^PD1^+^ T cell populations, as well as seven soluble immune molecules (i.e., IFNγ, sPDL2, sHVEM, sPD1, sGITR, sPDL1, and sCTLA4) associated with the clinical responses of mRCC patients, either modulated by TKI therapy or not. These results suggest an immunological profile of mRCC patients, which will help to improve clinical decision-making for RCC patients in terms of the best combination of strategies, as well as the optimal timing and therapeutic sequence.

## 1. Introduction

Renal cell carcinoma (RCC) represents 2–3% of cancer diagnoses in adults [1]. To date, nephrectomy remains the main therapeutic choice for most patients with localized disease; however, one-third of patients present metastatic disease at diagnosis and one-quarter of all patients could ultimately experience disease relapse. In the past decade, the prognosis of metastatic renal carcinoma (mRCC) has considerably improved due to the recent introduction of the vascular endothelial growth factor receptor–tyrosine kinase inhibitors (VEGFR–TKIs) and immune checkpoint inhibitors (ICIs). New synergistic combinations between TKIs and ICIs could increase the first line of therapeutic strategies in RCC. Although the recent improvements and advances in genomic sequencing and molecular characterizations have allowed an accurate definition of prognosis, predictive biomarkers are still needed to select the patients beneficiaries of the different therapeutic approaches. Diagnostic tools that pool biomarker data could help to tailor treatment strategies based on the biological and immunological parameters of the patient [2].

Indeed, it is well-known that immunological features can affect the prognosis of patients, but it has also been described in depth that target therapy presents several immunological effects. From this perspective, a dynamic immunological portrait of patients and cancer can influence not only the response to immunotherapy, but also the response to target therapy.

The question is how to increase the number of responsive patients to target and immunotherapy; therefore, it is necessary to understand the immunity underlying patients. The immune system represents, in fact, a key point for the clinical outcome of RCC patients, taking into account the potential prognostic values such as the tumor-infiltrating immune cells that create a microenvironment regulating cancer progression [3]. Furthermore, several immunosuppressive molecules, such as vascular endothelial growth factor (VEGF), characterize the microenvironment of this tumor with the ability to promote neo-angiogenesis and tumor growth as well as negatively impact immune response. VEGF signaling modulates T cell biology and function. Indeed, VEGF decreases T-cell progenitors in the thymus and differentiated T cells in the lymphoid organs and dampens their effector function. Furthermore, VEGF fosters immune-suppression by accumulation of regulatory T cells (Tregs) and contributing to T-cells exhaustion. Thus, while the neoangiogenic hallmark always represents a crucial pathway in RCC, making this tumor sensible to antiangiogenic therapies [4,5], also the immune system can be considered an off target for these therapies [6,7]. Indeed bevacizumab and sorafenib reverse the immunosuppressive effects of VEGF and restore the maturation of dendritic cells (DCs) [8,9]. Pazopanib improves DC differentiation and maturation and seems to modulate the CD137^+^ (4-1BB, a member of the TNF-receptor family) T-cell population [10]. Other TKIs, such as sunitinib, modulate immunosuppressive cells such as MDSC and Tregs [11,12].

Moreover, recent evidence demonstrates that several soluble immune molecules involved in immune regulation, such as soluble immune checkpoint-related proteins (sICs; i.e., es.sCTLA-4 and sPD1), can influence the development, prognosis and treatment of cancer [13]. These are functional proteins released by immune cells as alternative splice variants or by cleavage of membrane-bound proteins and can diffuse in serum [14,15]. However, only a few studies have evaluated the role of sICs in in the outcome of renal cancer.

The aim of our study was to evaluate the immunological effect of TKIs and the impact of the immune profile of patients in response to TKI therapy.

## 2. Results

### 2.1. Patient Characteristics

#### Study Population

The characteristics of patients are listed in Table 1. Twenty mRCC patients were enrolled. The median age at diagnosis was 56.5 years (range: 36–78 years); 15 (75%) patients were males and nine (45%) had a previous history of smoking. Clear cell RCC was the most represented histology (16 patients; 80%), followed by one case of clear cell carcinoma with sarcomatoid features, one case of chromophobe, and two cases with another histology. According to Fuhrman grading, nine patients (45%) were defined as G3, seven (35%) as G2, and four (20%) cases as unknown. Almost all patients in the study underwent nephrectomy (18 patients; 90%); 11 patients (55%) had metastatic disease at the first diagnosis of renal cancer. At diagnosis of metastatic disease, liver metastases occurred in four patients (20% of cases), whereas nodal, lung, bone, brain, and adrenal metastases occurred in eight patients (40%), 12 patients (60%), five patients (25%), three patients (15%), and one patient (5%), respectively. Overall, five patients were classified as poor risk according to their Metastatic Renal Cell Cancer Database Consortium (IMDC) scores, 10 patients as intermediate risk, and five patients as good risk. With regard to first-line treatment, eight (40%) and 12 (60%) patients received sunitinib and pazopanib, respectively. The toxicities to first-line therapy were in line with the treatment received. Most patients discontinued the first-line therapy due to progressive disease (16 patients; 80%); one patient (5%) stopped the treatment because of toxicity, while three patients (15%) remained on treatment. The first-line median progression-free survival (PFS) was 11 months (range: 1–31 months). Ten patients (50%) underwent second-line treatment with Nivolumab. Globally, second-line treatment was well tolerated; however, five patients (50%) stopped second-line treatment due to progressive disease and one patient due to toxicity, while in four patients, the treatment was ongoing at the last follow-up visit. The second-line median PFS was 4 months (range: 1–22 months). Of these patients, two received third-line therapy with cabozantinib. In one patient, the PFS was 8 months and treatment was discontinued for progressive disease, while in one patient, treatment remained ongoing. Responsive and non-responsive patients were considered on the basis of the first clinical revaluation and 3–4 months after beginning TKI treatment. Clinical and radiological outcomes were assessed as parameters to differentiate responsive and non-responsive patients. Tumor response was assessed every 3–4 months using immune-related Response Evaluation Criteria in Solid Tumors (i-RECIST). According to i-RECIST, responsive patients (R) were considered those who achieved complete or partial radiological response or a stable disease at the first radiological evaluation. Conversely, patients who experienced a radiological progression of disease or a clinically significant worsening of cancer-related symptoms were considered as non-responders (NR).

### 2.2. CD137^+^ T Cells Are Associated with the Response to TKIs in mRCC Patients

CD137 is a co-stimulatory molecule expressed on activated T cells, and the engagement with its ligand contributes to enhancing the proliferation and effector functions of lymphocytes, preventing apoptosis [16]. It is considered a bonafide marker of recently activated tumor-reactive T cells. 

Figure 1A shows the expression of CD137 molecule on CD3^+^, CD8^+^, and CD4^+^ T cells in responsive and non-responsive patients before treatment (T0) and during treatment (>T0). At T0, responsive patients had a significantly higher percentage of CD3^+^CD137^+^ T cells (2.7% ± 0.92%) compared to non-responsive (0.9% ± 0.87%) (*p* = 0.003), which was also maintained during TKI treatment (%CD3^+^CD137^+^: 2.6% ± 0.78% in responsive patients vs. 0.67% ± 0.4% in non-responsive patients; *p* = 0.0001). In particular, CD137 expression was associated with the CD8^+^ T cell subpopulation. In fact, at T0, the expression of CD137 on CD8^+^ T cells was significantly higher in responsive patients (2.02% ± 0.7%) compared to non-responsive patients (0.6% ± 0.5%) (*p* = 0.001). The same significant trend was observed during TKI treatment (CD8^+^CD137^+^ subpopulation was 1.91% ± 0.75% in responsive patients vs. 0.43% ± 0.25% in non-responsive; *p* = 0.0008). Instead, no significant differences were obtained for CD4^+^ T-cell subpopulation (%CD4^+^CD137^+^ at T0: 0.6% ± 0.2% in responsive patients vs. 0.27% ± 0.18% in non-responsive, *p* = 0.28; at >T0: 0.87% ± 0.28% in responsive vs. 0.23% ± 0.08% in non-responsive, *p* = 0.18).

These results show that CD137^+^ T cells could represent a possible biomarker that is able to identify patients that could clinically benefit from TKI treatment.

The Kaplan–Meier survival curves for patients with high and low concentrations of CD8^+^CD137^+^ T cells are shown in Figure 1B. During treatment with TKI, the median survival times were 12 months in the group with a low concentration of CD8^+^CD137^+^ T cells and undefined in the group with a high concentration CD8^+^CD137^+^ T cells (*p* = 0.04, log-rank test). The same trend was observed at baseline, despite the fact that the difference between high and low concentrations of CD137 T cells was not statistically significant. These data suggest that the maintenance of CD8^+^ CD137^+^ T cells in circulation is associated with the duration of the response to TKIs.

The expression of PD1 molecules on the CD3^+^CD137^+^ T-cell population was also analyzed (Figure 1C). It was observed that during TKI treatment, responsive patients experienced a significant downregulation of PD1 expression (*p* = 0.02). Moreover, at >T0, PD1 resulted significantly higher in non-responsive compared to responsive patients (27.9 ± 3.4 vs. 16 ± 3.2, respectively; *p* = 0.04).

The higher percentage of PD1^+^ on CD3^+^CD137^+^ T cells in non-responsive patients could suggest that these patients could possibly benefit from an anti-PD1 therapy already at the time of evaluation.

Regulatory T cells and other exhaustion markers were analyzed, but no difference was observed between patients.

### 2.3. TKI Treatment Modulates Soluble Immune Molecules

In order to evaluate the impact of TKI treatment in the release of soluble immune molecules in mRCC patients, the levels of immune checkpoint-related proteins and inflammatory cytokines were evaluated in the sera of mRCC patients before (T0) and during TKI therapy (>T0). It was recently demonstrated that the soluble isoforms of the checkpoint receptors can contribute to immune regulation, representing putative biomarkers for tumor outcome and patient stratification [17]. Moreover, much evidence has demonstrated that these molecules are involved in positive or negative immune regulation and that changes in their plasma levels affect the development, prognosis, and treatment of cancer [13].

Figure 2A shows that TKI treatment in mRCC patients modulates several sICs. In particular, the concentration of sPDL2 significantly decreased during TKI therapy (7842.5 ± 2865 pg/mL for T0 vs. 4989 ± 4462 pg/mL for >T0; *p* = 0.02). Similar results were observed for sHVEM (4085.5 ± 3388 pg/mL for T0 vs. 1777 ± 1578 pg/mL for >T0; *p* = 0.01). It was shown that the high concentration of sHVEM seems to contribute to tumor development and progression [18]. Moreover, the results indicate that TKI treatment also affects the release of sPD1 and sGITR, decreasing the concentration of both molecules between T0 and >T0 (sPD1: 561.5 ± 431 pg/mL for T0 vs. 238 ± 176 pg/mL for >T0, *p* = 0.02; sGITR: 548 ± 425 pg/mL for T0 vs. 214 ± 212 pg/mL for >T0, *p* = 0.01).

The correlations between the fold-changes (>T0/T0) of soluble immune checkpoint molecules were also calculated and are shown in Appendix A. The fold-change of sPD1 was positively correlated with that of sGITR (*p* = 0.009, *r* = 0.65) and sHVEM (*p* = 0.002, *r* = 0.59). Additionally, a positive correlation between the fold-change of sHVEM and sGITR (*p* = 0.0009, *r* = 0.76) was also found, while no correlation was obtained for sPDL2.

When the association between these soluble molecules and clinical responses was evaluated (Figure 2B), sPD-L2 resulted in the significant modulation of unique soluble immune checkpoint-related proteins in mRCC responsive patients (R) during TKI treatment (sPDL2: 8855 ± 3985 pg/mL for T0 vs. 5057 ± 4243 pg/mL for >T0, *p* = 0.01). No significant modulation was obtained in non-responsive patients (NR). On the other hand, it was recently demonstrated that sPDL2 is the strongest predictor of recurrence in ccRCC; patients with a high level of sPDL2 had, in fact, a significantly increased risk of recurrence [19].

These data demonstrate that TKIs impact the release of immune molecules, suggesting their possible role in the clinical outcome of mRCC patients.

The results were independent of the TKI administered and no significant data were obtained for other soluble factors tested (cytokine and checkpoint related proteins; Appendix A).

### 2.4. TKI Responsive Patients Have Low Levels of Serum IFNγ

To analyze the contribution of immune cytokines to the response to treatment, the sera of mRCC responsive and non-responsive patients were analyzed at T0 and during TKI therapy. Figure 3A shows that before beginning TKI therapy, those mRCC patients that would benefit from treatment had a significantly lower concentration of IFNγ compared to non-responsive patients (27.47 ± 8.5 pg/mL for R vs. 515.8 ± 210.6 pg/mL for NR; *p* = 0.007). The same significant trend was found for >T0 (48.74 ± 21.24 for R patients vs. 267.8 ± 77.12 for NR; *p* = 0.002). These data show that low levels of IFNγ correlates with response to TKI treatment. IFNγ plays a key role in antitumor immune responses in the elimination stage of the immunoediting paradigm. However, recent evidence suggests that IFNγ may also play a significant role in promoting tumorigenesis [20].

To determine whether mRCC patients treated with TKIs derive survival benefit based on IFNγ levels, survival rates were examined. Figure 3B shows that IFNγ predicts, at baseline, the duration of the response to TKI treatment in mRCC patients. Patients with low levels of IFNγ (<65 pg/mL) had a longer duration of response to TKI therapy compared to patients with higher levels (>65 pg/mL). The average time of the duration of the response was undefined vs. 7 months (*p* = 0.04), and no significant correlation was observed during treatment (>T0).

### 2.5. Upregulation of sPDL1 and sCTLA4 in Non-Responsive Patients During TKI Treatment

The serum levels of other cytokines and soluble checkpoint-related proteins were evaluated according to the response to therapy at T0 and >T0. Among the molecules analyzed, sPDL1 and sCTLA4 resulted statistically significant. In particular, as shown in Figure 3C, the average concentration values in the serum obtained at T0 for sPDL1 and sCTLA4 highlighted a trend of a higher release of these molecules in non-responsive patients. This difference was statistically significant during TKI treatment: sPDL1 levels in responsive patients were 56.25 ± 36.5 pg/mL vs. 146.5 ± 122.3 pg/mL for non-responsive patients (*p* = 0.03); sCTL4 levels were 281.6 ± 133 pg/mL for responsive patients compared to 616.4 ± 330.3 pg/mL for non-responsive patients (*p* = 0.008). These data suggest a possible higher circulating immunosuppressive status in mRCC patients that would not benefit from TKI therapy.

## 3. Discussion

Angiogenesis plays a key role in RCC tumorigenesis and progression, directing the immune system through the abnormal formation of tumor vessels and the promotion of an immunosuppressive tumor microenvironment. Therefore, antiangiogenic treatments remain a valid therapeutic option in selected patients, since they modulate immune responses [10,21,22]. This activity is essential to enhancing the performance of immunotherapy agents, which have shown promising treatment outcomes also for advanced RCC [23]. Immune checkpoint inhibitors combined with TKIs will become a new standard of care in treatment-naive patients with advanced RCC [6]. However, not all patients benefit from immune checkpoint therapy and, at present, no effective biomarkers can be included in the therapeutic algorithm, despite large research efforts. Thus, the identification of reliable predictive factors is necessary. Only a few studies have investigated the role of immune cells and circulating immune molecules in RCC, especially in metastatic cancer patients. In this study, we observed, for the first time, that TKIs are able to modulate soluble immune checkpoint-related proteins. Moreover, we identified an association between circulating biomarkers and the response to TKI treatment. In particular, we identified that CD3^+^CD8^+^CD137^+^ T cells are a population of activated T lymphocytes significantly more expressed in responsive patients, both at baseline than during TKI treatment, suggesting that CD137 could represent a predictive biomarker of response to TKI. The CD137 receptor is considered a biomarker of tumor-reactive cells. It has been demonstrated that signaling through CD137 induces the activation of CD8^+^ T cells in a CD28-independent manner, enhancing T-cell survival, promoting their effector function, and favoring memory differentiation [24,25]. The results obtained in our study are in line with that observed in a previous study conducted on a limited number of mRCC patients, where we identified modulations occurring in the immune T cell repertoire of mRCC during TKI treatment. Among the different biomarkers tested, we were able to detect a CD137^+^ T-cell subset in mRCC arising during pazopanib treatment [10]. In this study, we further observed that this population correlates with the response to TKI therapy. In fact, we obtained an increase in CD3^+^CD137^+^ T cells at baseline in those mRCC patients who benefited from TKI treatment. Moreover, we observed that CD3^+^CD137^+^ T lymphocytes are mainly cytotoxic T cells and that the maintenance of CD8^+^CD137^+^ in circulation is associated with the duration of response to treatment, suggesting that CD137 could represent a promising target for antitumor immune activation strategies. Numerous studies have demonstrated that although both activated CD4^+^ and CD8^+^ T cells express CD137, signals through CD137 are more biased toward CD8^+^ T cells, both in vitro and in vivo [26,27,28]. Interestingly, the in vivo administration of agonistic anti-CD137 antibody promotes CD8^+^ T-cell expansion, providing protection against several diseases, including cancer [29,30]. In fact, CD137 has previously been shown to be important in positively regulating effector T-cell responses in cancer [31,32]. Freeman et al. recently observed that, in tumor types heavily infiltrated with CD8^+^ T cells, CD137 is associated with increased CD8^+^ T-cell effector function and improved patient survival [33].

In this study, we also observed the expression of PD1 on CD137^+^ T cells, particularly in non-responsive patients during TKI therapy. A previous report showed the presence of a rare population of a CD8^+^CD137^+^PD1^+^ T cell subset in lung cancer patients [24]. Moreover, it was demonstrated that the co-expression of PD-1 and CD137 on tumor-infiltrating lymphocytes (TILs) contributes to the synergistic effects of the combination of anti-PD-1-blocking agents and CD137 agonists [34]. These data suggest that a subpopulation may potentially exert a critical antitumor effect through combination immunotherapy with anti-PD1-blocking agents and CD137 agonists, opening up possible new therapeutic strategies in the management of advanced RCC patients. On the other hand, immune checkpoints play important roles in immune regulation, and blocking immune checkpoints on the cell membrane has proven to be an effective strategy in the treatment of cancer [13]. However, the influence of soluble receptors and ligands on immune modulation and cancer outcome has not been studied. To date, only few studies have investigated the contribution of soluble factors to the clinical efficacy provided by ICIs; this has led to the incorrect assumption that these antibodies act only at the level of membrane-based interactions [17]. Soluble receptors and ligands, which are part of a family that includes full-length receptors and ligands, are produced by mRNA expression or by the cleavage of membrane-bound proteins and are found free in the plasma. Recent studies indicate that soluble isoforms of immune checkpoint receptors are centrally involved in immune regulation and, to date, only few studies have examined the association between soluble immune checkpoint-related proteins and cancer outcomes [35].

Here, we identified soluble immune checkpoint-related proteins and inflammatory cytokines modulated by TKI therapy and associated with the clinical outcomes of mRCC patients. It is well-known that TKIs affect the immune system [6]. Antiangiogenic therapies contrast the immunosuppressive effects induced by angiogenic factors, increasing the tumor infiltration of mature DCs and effector T cells and decreasing the infiltration of immunosuppressive cells, mainly regulatory T cells and myeloid-derived suppressor cells [8,11,12,36,37,38]. Nevertheless, the impact of TKI treatment on soluble immune checkpoint-related inhibitors and the possible role induced by these factors in response or resistance to treatment has not yet been elucidated in mRCC. Recently, it was demonstrated that sorafenib induces changes in soluble checkpoint protein levels in patients with advanced hepatocellular carcinoma [15]. Here, we demonstrated that TKIs modulate soluble ICs (i.e., sPDL2, sHVEM, sPD1, and sGITR) and we identified sPDL2 as a unique biomarker significantly downregulated in responsive patients during TKI treatment. This finding is supported by previous work describing sPDL2 as the most significant predictive biomarker of recurrence risk in ccRCC [19]. The high expression of PDL2 in tumors is also correlated with decreased cancer-free survival in RCC patients [39]. Moreover, high levels of sPDL2 are correlated with poor survival post-CAR T cells infusion in relapsed and refractory B-cell lymphoma and leukemia patients [40]. Therefore, we speculate that sPDL2 represents a biomarker correlated in response to TKI treatment, although these findings warrant further confirmation. Moreover, our results indicate that changes in soluble ICI during TKI treatment are positively correlated to one another, in contrast to sPDL2, which seems to be an independent factor. According to several studies that have reported the role of soluble checkpoint molecules in the promotion and progression of cancer, downregulating immune activation [18], here we demonstrated that IFNγ, sPDL1 and sCTLA4 play important roles in regulating the response to TKI treatment. Interestingly, we observed that low levels of IFNγ correlated with the response to TKI therapy, both at baseline and after 3–4 months after starting treatment. Moreover, the low levels of IFNγ at baseline seems to be associated with a better response to TKI treatment in terms of the duration of the response. Indeed, early studies on IFNγ and cancer biology established its role as an antitumor cytokine; however, now it is known that this cytokine can have a dual role in shaping the outcome of cancer [41]. A lot of genes induced by IFNγ are, in fact, involved in cancer cell immune evasion, such as PDL1, PDL2, CTLA4, and IDO [42,43]. It was shown that IFNγ promotes the expression of PDL1 and PDL2, both on tumor cells and on immune infiltrating cells, and suppresses the effector function of tumor-specific T cells or NK cells through interaction with the immune inhibitory receptor PD1 [44,45]. Moreover, our data show that sPDL1 is associated with poor response to TKI treatment, confirming the predictive role of this molecule for poor prognosis and increased risk of death in mRCC [46,47,48]. sCTLA4 was the other soluble protein that showed a significant correlation with failure to response during TKI treatment. It is possible that the increase in sPDL1 and sCTLA is linked to the high levels of IFNγ, but this is a mechanism that needs to be evaluated in a larger number of patients. Recently, IFNγ-induced PD-L1/2 expression was also referred to as a mechanism of adaptive immune resistance to immune checkpoint therapy [49].

## 4. Materials and Methods

### 4.1. Patient Selection

This was a multicenter, prospective, observational study. Twenty consecutive patients affected by metastatic renal carcinoma, referred to three Italian oncology units (i.e., Policlinico Umberto I Hospital, Sapienza University; A. Gemelli Hospital, Cattolica University; San Camillo Forlanini Hospital) and treated with at least one line of treatment, were enrolled. Blood samples were collected at different time points: before starting and during therapy (i.e., at the first clinical revaluation and 3–4 months after beginning treatment). The clinical and survival data of identified patients were retrieved from clinical records. A specific database, including the following clinical and pathological features for each patient, was built: age, sex, smoking status, histology, Fuhrman grading, date of diagnosis of metastatic disease, date of nephrectomy, IMDC score, first-line treatment, date of progression to first-line treatment, toxicities related to first-line treatment, second-line treatment, date of progression to second-line treatment, toxicities related to second-line treatment, and third-line treatment. Written informed consent was obtained from all patients. The study was approved by the Ethics Committee of Policlinico Umberto I (Ethical Committee Protocol, RIF.CE: 4181).

### 4.2. PBMC (Peripheral Blood Mononuclear Cells) Purification and Sera Collection

PBMCs were isolated from the peripheral blood (40 mL) of mRCC patients before (T0) and during TKI therapy (>T0, at first clinical revaluation) using Ficoll–Hypaque gradient (1077 g/mL; Pharmacia LKB). At the same time, sera were collected and stored at −80 °C until use.

### 4.3. Immune Phenotype

A multi-parametric analysis by flow cytometry was conducted to evaluate various T-cell subsets and function as the combined expression of the following markers:T cell exhaustion/activation: Anti-CD3-APC-H7/CD8-PerCp-Cy5.5/CD137-PeCy7/PD1-PE/C TLA4-APC/Tim3-BB515;T regulatory cells: Anti-CD4-APC-H7/CD25-PE/CD45RA-BB515/FoxP3-Alexa647.

Flow cytometric analysis was performed using a FACSCanto flow cytometer running FACS Diva data acquisition and analysis software (version 8.0.2, BD Biosciences, San Diego, CA, USA). In Appendix A are listed the catalog numbers and clones of the antibodies used in this study.

### 4.4. Inflammatory Cytokine, Chemokine and Soluble Checkpoint Inhibitor Detection

The sera from mRCC patients were assayed to quantify cytokines and soluble checkpoint molecules using the ProcartaPlex Human Inflammation Panel (20 Plex, catalog number EPX200-12185-901; sE-Selectin; GM-CSF; ICAM-1/CD54; IFN alpha; IFN gamma; IL-1 alpha; IL-1 beta; IL-4; IL-6; IL-8; IL-10; IL-12p70; IL-13; IL-17A/CTLA-8; IP-10/CXCL10; MCP-1/CCL2; MIP-1alpha/CCL3; MIP-1 beta/CCL4; sP-Selectin; TNF alpha) (eBioscence, Vienna, Austria) and the Human Immuno-Oncology Checkpoint 14-Plex ProcartaPlex Panel 1 (catalog number EPX14A-15803-901; BTLA; GITR; HVEM; IDO; LAG-3: 47; PD1; PD-L1; PD-L2; TIM-3; CD28; CD80; CD137; CD27; CD152) (eBioscence). Samples were measured using Luminex 200 platform (BioPlex; Bio-Rad, Bio-Rad, Hercules, CA, USA) and data, expressed in pg/mL of protein, were analyzed using Bio-Plex Manager Software (version 6.1, Bio-Rad).

### 4.5. Statistical Analysis

Descriptive statistics (i.e., average and standard deviation) were used to describe the various data. Student’s paired and unpaired *t*-tests were used to compare two groups. Statistical significance was indicated when the *p*-value was less than 0.05.

Keplan–Meier analysis and log-rank test were used to evaluate the percentage of survival related to the duration of response. Correlations of fold-changes in levels of two proteins were assessed through Spearman’s rank correlation test. A *p*-value of <0.05 was considered statistically significant.

## 5. Conclusions

Our data identified new circulating biomarkers, both cells and molecules, that are associated with TKI treatment/response in mRCC and that allow to characterize responsive and non-responsive patients (Table 2). Subsequent validation studies will be performed to validate these markers in a network medicine framework and to test their predictive value for treatment outcomes both in TKI- and immunotherapy-treated patients. These new developments in the research of biomarkers could considerably improve clinical decision-making for RCC patients. Defining the best combination for each patient, as well as the optimal therapeutic sequence, will be essential to guide treatment decisions in clinical practice. In conclusion, the immune-modulating effects of TKI open the way to new therapeutic strategies for mRCC and other cancers, suggesting variations in the administration timing of treatment and new possible combinations with other TKIs, ICIs, or of immune agents, including cancer vaccines and immunostimulatory agents.

## Figures and Tables

**Figure 1 cancers-12-02620-f001:**
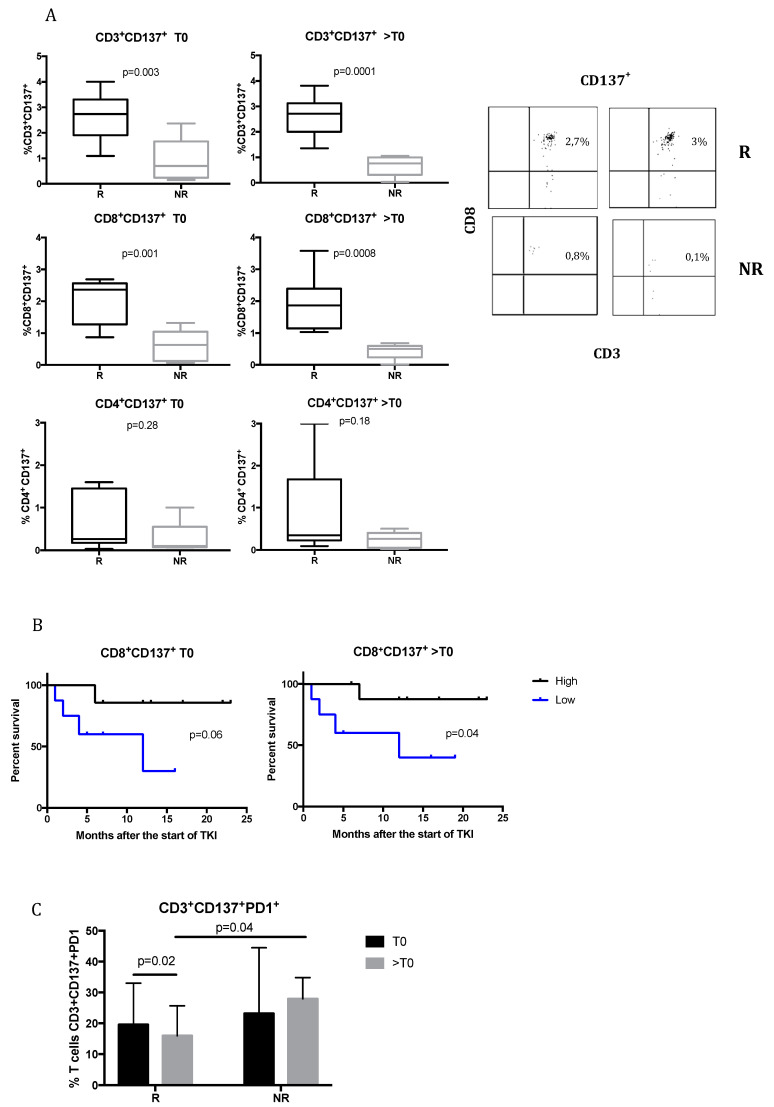
(**A**) Immune cell subpopulations were evaluated using flow cytometry and analyzed by FACSDiva Software. To analyze the CD137^+^ T cells, lymphocytes were first gated on FSC-A and SSC-A, and then the CD3^+^ T-cell subpopulation was selected from the lymphocytes. CD3^+^CD137^+^ T cells were then selected and analyzed for CD4 and CD8. The results are shown as percentages of CD3^+^CD137^+^, CD8^+^CD137^+^ and CD4^+^CD137^+^ T cells in responsive (R) and non-responsive (NR) patients at baseline (T0) and during tyrosine kinase inhibitor (TKI) treatment (>T0). The dot plot analysis of the CD3^+^CD8^+^CD137^+^ T lymphocytes is shown in the right of panel A. The results are representative of one R patient and one NR metastatic renal carcinoma (mRCC) patient. (**B**) Survival analysis at baseline and during treatment of mRCC patients treated with TKI. At T0, survival analysis of the mRCC patients was conducted, comparing those with greater than 1.4% of CD8^+^CD137^+^ T cells to those with less or equal to 1.4%. During TKI therapy (>T0), a survival curve was calculated using the value of 1.3% to distinguish high and low percentages of CD8^+^CD137^+^ T cells. Log-rank tests were used to compare the survival between two groups. (**C**) Expression of PD1 molecules on the CD3^+^CD137^+^ T lymphocytes. The results are reported as percentages of PD1 normalized on CD3^+^CD137^+^ T cells in R and NR patients a T0 and during TKI therapy (>T0). Statistical significance was determined by a Student’s unpaired *t*-test. A *p*-value of <0.05 was considered statistically significant.

**Figure 2 cancers-12-02620-f002:**
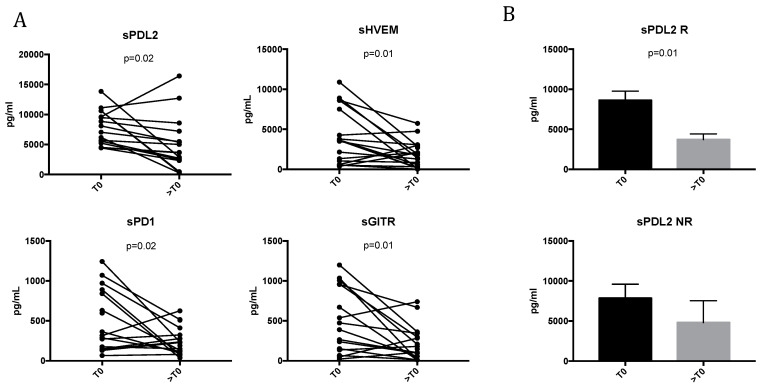
Changes in the soluble immune checkpoint-related proteins during TKI therapy in mRCC patients. (**A**) Analysis of soluble immune checkpoint-related proteins levels (i.e., sPDL2, sHVEM, sPD1, and sGITR) in patients with mRCC at baseline (T0) and after 3–4 months of TKI treatment (>T0). The proteins were analyzed by Luminex multiplex assay and the results are reported as the concentration (pg/mL) of soluble checkpoint inhibitors present in the serum of mRCC patients. (**B**) sPDL2 levels in the serum of mRCC responsive (R) and non-responsive (NR) patients analyzed at T0 and >T0. sPDL2 resulted in the only significantly modulated molecule associated with response to TKI treatment. Statistical significance was determined by a Student’s paired *t*-test, and a *p*-value < 0.05 was considered statistically significant.

**Figure 3 cancers-12-02620-f003:**
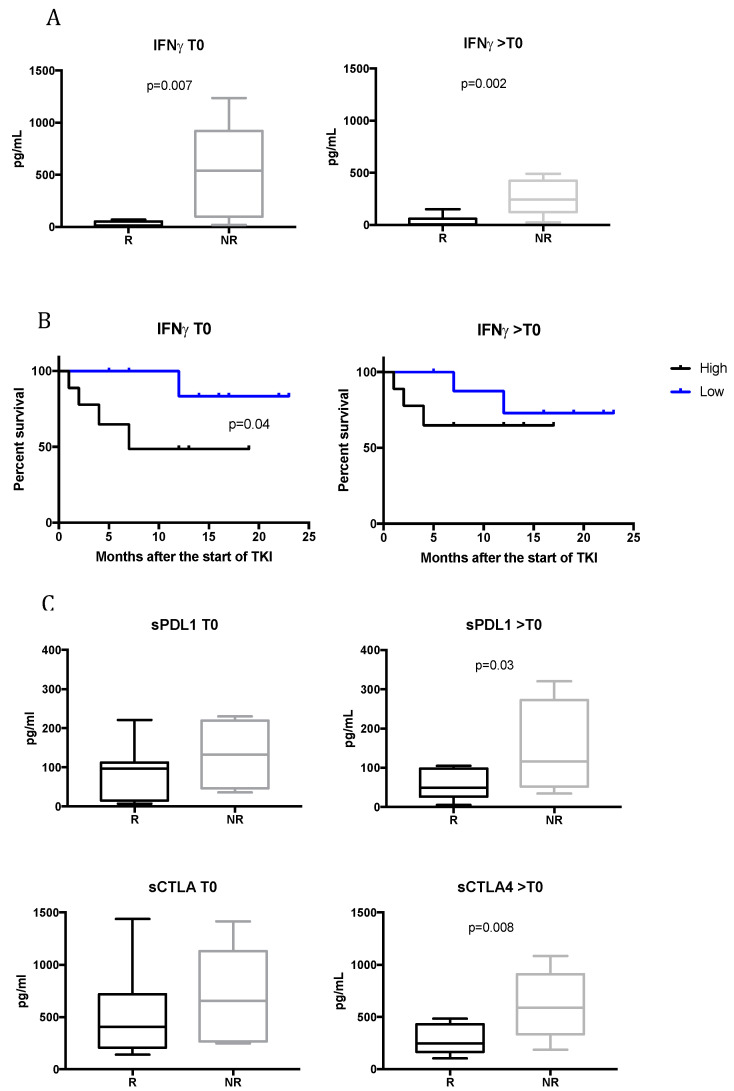
Profiling of levels of immune molecules at baseline and during TKI treatment in responsive (R) and non-responsive (NR) patients. (**A**) Box plots of IFNγ levels in R and NR mRCC patients at T0 and >T0. The lines in the boxes show the median values. The error bars show the minimum and maximum values. (**B**) Survival curve analysis of the mRCC patients at baseline and during TKI treatment according to the levels of IFNγ. For T0, the median value considered for patients belonging to the high-concentration group was >65 pg/mL, while for those belonging to the low-concentration group was ≤65 pg/mL. For the analysis of survival during TKI treatment, the median value of IFNγ levels used to dichotomize patients was >59 pg/mL for the high-concentration group and ≤59 pg/mL for the low-concentration group. A log-rank test was used to compare the survival between two groups. (**C**) Box plots of sPDL1 and sCTLA4 at baseline and 3–4 months after the start of TKI therapy (>T0). A Student’s unpaired *t*-test was used to compare R vs. NR patients and a *p*-value < 0.05 was considered statistically significant.

**Table 1 cancers-12-02620-t001:** Clinical and pathological characteristics and treatment.

Characteristic	All Patients (*N* = 20) (100%)
Age (years)	56.5
Median Age (range)	(36–78)
Gender	
Male	15 (75)
Female	5 (25)
Risk Factors	9
Smoking history (SH)	(45)
Histology	
Clear cell carcinoma	16 (80)
Other	4 (20)
Fuhrman grading	
G2	7 (35)
G3	9 (45)
Unknown	4 (20)
Metastatic site at diagnosis	
Liver	4 (20)
Nodal	8 (40)
Lung	12 (60)
Bone	5 (25)
Brain	3 (15)
Adrenal	1 (5)
IMDC score	
Poor risk	5 (25)
Intermediate	10 (50)
Good risk	5 (25)
I-line treatment	20
Sunitinib	8 (40)
Pazopanib	12 (60)
II-line treatment	10 (50)
Nivolumab	10 (100)
III-line treatment	2
Cabozantinib	2

**Table 2 cancers-12-02620-t002:** Circulating biomarkers modulated in mRCC patients.

mRCC Patients	Baseline	During TKI Treatment
Responsive patients	High CD3^+^CD8^+^CD137^+^ Low PD1 on CD137^+^ T cells Low IFNγ	High CD3^+^CD8^+^CD137^+^ Low IFNγ Low sPDL2
Non-responsive patients	Low CD3^+^CD8^+^CD137^+^ High IFNγ	Low CD3^+^CD8^+^CD137^+^ High PD1 on CD137 T cells High IFNγ High sPDL1 High sCTLA4
sICs modulated by TKI	sPDL2, sHVEM, sPD1, sGITR	

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
