# Peer review of "Exploratory Pilot Study of Circulating Biomarkers in Metastatic Renal Cell Carcinoma"

_cancers, 2020, doi:10.3390/cancers12092620_

Round 1
Reviewer 1 Report
In this manuscript, Zizzari et al evaluates the immune status of mRCC patients in an effort to identify a combination of immunological biomarkers relevant to improve mRCC patient’s outcome. The author has profiled 20 mRCC patients for their immune cell status and other immune checkpoint related proteins with or without TKI therapy. In short, the aim of the study is clear, however, the reviewer has concern about the significant outcome of the study. The current data does not satisfactorily addressed the conclusion made in the manuscript.
Major concerns,
Line 77, Study population- there was no data provided how the study populations are characterized as responsive or non-responsive patients? What are the parameters considered to differentiate these groups?
Line 97; the author stated “Responsive and not responsive patients are considered on the basis of the first clinical revaluation, 3-4 months after the beginning of TKi treatment”. However, fig.1 the graphs compared R and NR within baseline T0 or >T0. It is must to compare the R group at T0 to R group at >T0 to find the effect of TKI therapy. From the graph I could not find a significant effect of the TKI therapy on both R or NR groups, as the T cell frequency is same before and after therapy in both R or NR groups.
Line 107 to 128; CD137 is a member of the TNFR family and is rapidly expressed on activated CD8+ T cells and also on activated CD4+ T cells with lower levels. However, the flow gating used to analyze T cell subsets are not appropriate as CD137 should be analyzed on CD4 or CD8 T cells, but it was reversed on the current study. Also, I do not see any data related to CD4 cells, there are only CD3+ and CD8+ cells were analyzed. At baseline, both R and NR groups have similar T cell frequencies with or without TKI therapy.
Fig.3C & line 141-148, the data of PD1 expression on T cells and the conclusion drawn are not convening. Even though the frequency of PD1+ cells in the NR groups is higher, the absolute total cells in these NR groups are much lower (dot plot Fig1A) which further impact the conclusion.
Fig.2 this is an interesting data that the TKI therapy effectively reduce the several soluble proteins that may involve in immune regulation. However, describing these soluble factors as a biomarker need strong evidence.
Fig.3; Again there is no rational in comparing the R and NR groups at T0 or >T0, which may not tell the TKI therapy worked or not.
Line 347, the manuscript does not have any proliferation assay data.
Line 364, typos
Author Response
Response to Reviewer 1 Comments
Point 1: Line 77, Study population- there was no data provided how the study populations are characterized as responsive or non-responsive patients? What are the parameters considered to differentiate these groups?
Response 1: We thank the reviewer for this point which is missing in this final version. In fact this is a mandatory point that was missed in the submitted manuscript. We have inserted the following sentence, line 110-116: “Clinical and radiological outcomes were assessed as parameters to differentiate responsive and non-responsive patients. Tumor response was assessed every 3–4 months using immune-related Response Evaluation Criteria in Solid Tumors (i-RECIST). According to i-RECIST, we considered as responsive patients (R) those who achieved complete or partial radiological response or a stable disease at the first radiological evaluation. Conversely, patients who experienced a radiological progression of disease or a clinical significant worsening of cancer related symptoms were considered as non-responders (NR).
Point 2: Line 97; the author stated “Responsive and not responsive patients are considered on the basis of the first clinical revaluation, 3-4 months after the beginning of TKi treatment”. However, fig.1 the graphs compared R and NR within baseline T0 or >T0. It is must to compare the R group at T0 to R group at >T0 to find the effect of TKI therapy. From the graph I could not find a significant effect of the TKI therapy on both R or NR groups, as the T cell frequency is same before and after therapy in both R or NR groups.
Response 2: As the reviewer correctly observes, we have not found a significant effect of TKi therapy comparing R group at T0 to R group at T>0 (or NR group), even if NR patients have an higher decreasing trend of CD3+CD137+ T cell frequency after therapy: CD3CD137 NR (T0: 0,86% ± 0,38; >T0 : 0,44% ± 0,16); CD3CD137 R (T0: 2,65% ± 0,28 ; >T0 2,55% ±0,23).
However the main objective of Fig.1A is to highlight that the CD3+CD137+ T cell population, and in particular CD8+CD137+, can be a useful circulating biomarker able to distinguish R from NR patients both at baseline and >T0, demonstrating that patients who will be responsive to TKi treatment have a % of these population significantly higher even before starting TKi therapy. Moreover experimental observations are confirmed by clinical data. Fig.1B shows in fact as the presence of CD8+CD137+ is associated with better percentage of survival (100% of NR patients are placed in the blue curve, with low levels of CD8+CD137+ T cells, instead the 75% of R patients in the black one).
Point 3: Line 107 to 128; CD137 is a member of the TNFR family and is rapidly expressed on activated CD8+ T cells and also on activated CD4+ T cells with lower levels. However, the flow gating used to analyze T cell subsets are not appropriate as CD137 should be analyzed on CD4 or CD8 T cells, but it was reversed on the current study. Also, I do not see any data related to CD4 cells, there are only CD3+ and CD8+ cells were analyzed. At baseline, both R and NR groups have similar T cell frequencies with or without TKI therapy.
Response 3: In the current study we used the same gating strategy described and published by ourselves on Cancer Immunology Research (Zizzari IG at al. 2018). We have first gated the lymphocytes on FSC-A and SSC-A, then we gated on lymphocytes the CD3+ T cells. On CD3+ T cells we have selected the CD137+CD3+ T cells and finally on CD3+CD137+ we selected CD4+ and CD8+ CD137+. We considered the lymphocytes CD3+CD8- as CD4+T cells. Moreover, as correctly suggested by the reviewer, data related to CD4 T cells were added in Fig 1A (Line 146) and described in the results, line 133-136: “Instead, no significant differences were obtained for CD4+ T cell subpopulation (%CD4+CD137+ at T0: 0.6% ± 0.2% in responsive patients vs. 0.27% ± 0.18% in non-responsive, p=0.28; at >T0: 0.87% ± 0.28% in responsive vs. 0.23% ± 0.08% in non-responsive, p=0.18). We have also corrected the legend of the figure (Line 152)
Point 4: Fig.3C & line 141-148, the data of PD1 expression on T cells and the conclusion drawn are not convening. Even though the frequency of PD1+ cells in the NR groups is higher, the absolute total cells in these NR groups are much lower (dot plot Fig1A) which further impact the conclusion.
Response 4: Our interpretation of this data is that NR patients have not only a lower number of CD3+CD137+ T cells compared to R group (dot plot Fig1A), but these cells also express more PD1 in proportion, specially during TKI therapy (>T0), suggesting a more exhausted profile.
Point 5: Fig.2 this is an interesting data that the TKI therapy effectively reduce the several soluble proteins that may involve in immune regulation. However, describing these soluble factors as a biomarker need strong evidence.
Response 5: As we discuss in the manuscript, this is an exploratory study and we are acknowledged that future studies are necessary to validate these markers. However despite the small patient/sample size statistically significant differences are obtained and this support us to perform a study with a larger number of patients.
Point 6: Fig.3; Again there is no rational in comparing the R and NR groups at T0 or >T0, which may not tell the TKI therapy worked or not.
Response 6: As the reviewer observes, Fig. 3 does not compare R and NR groups at T0 or >T0. This comparison was shown in fig 2, where among the T cells, cytokines/chemokines and soluble check point inhibitors analyzed, we reported the markers significantly modulated by TKi treatment. Instead Fig.3 shows which of the soluble factors can discriminate a R patient from a NR patient both at baseline and during TKI therapy.
Point 7: Line 347, the manuscript does not have any proliferation assay data.
Response 7: Absolutely correct, we changed the Line 379 and deleted line 381
Point 8: Line 364, typos
Response 8: Thank you very much, we corrected (line 398)

Reviewer 2 Report
This manuscript describes the expression levels of soluble immune factors and T cell subsets associated with renal cell carcinoma and response to TKI inhibitors of VEGFR as potential biomarkers of response.
The manuscript seems technically sound, however, is limited by the small patient/sample size. This is acknowledged by the authors, nevertheless some statistically significant differences are obtained linking expression of particular molecules to response. In the case of increased CD3+/CD137+T cells this was observed in TKI 'responsive' patients both pre-treatment (T0) & 3-4 months after TKI treatment (T>0). Other differential effects were observed for IFNgamma, SPDL2 and other potential biomarkers - effects are summarised in Table 2.
Overall, this is a pilot study with some interesting findings that require a much larger patient study for confirmation/validation of key findings given the very small sample size. However, the manuscript appears generally sound for dissemination of these initial findings to the wider scientific community to facilitate such a study.
Minor points:-
1) Expand more on the immunosuppressive role of VEGF signalling in the introduction.
2) Manuscript requires careful proofreading by a native speaker of English. e.g. some improvement in grammatical English is required in places e.g. line 22-23; Line 67 – replace evidences with evidence
3) Title – mRCC – use full definition rather than abbreviation
4) Abstract (line 18)– replace abbreviations ICIs and TKI with full definition
5) Line 26 – typo
6) ‘Pts’ – non-standard abbreviation for patients – replace throughout
7) Replace not responsive patients with non responsive patients
8) Line 318-19 – unclear, rephrase
9) For Box & Whisker plots show individual data points
10) 8 patients recieved sunitinib and 12 patients received pazopanib - the latter is of broader TKI selectivity - the authors should comment on whether any differences in any of the putative biomarkers were obtained between the two different TKIs, and if so why this may be
Author Response
Response to Reviewer 2 Comments
Point 1: Expand more on the immunosuppressive role of VEGF signalling in the introduction.
Response 1: We agree with the request of the reviewer and have integrated the text as follow: Page 2, lines 62-69 “….Furthermore several immune suppressive molecules, such as VEGF, characterize the tumor microenvironment of this tumor with the ability to promote neo-angiogenesis and tumor growth as well as negatively impact immune response. VEGF signaling modulates T cell biology and function. Indeed, VEGF decreases T cell progenitors in the thymus and differentiated T cells in the lymphoid organs and dampen their effector function. Furthermore, VEGF fosters immune-suppression by accumulation of regulatoy T cells (Tregs) and contributing to T cell exhaustion. Thus, while the neoangiogenic hallmark always represents a crucial pathway in renal cell carcinoma making this tumor sensible to antiangiogenic therapies [4,5], also the immune system can be considered an off target for these therapies [6,7]”
Point 2: Manuscript requires careful proofreading by a native speaker of English. e.g. some improvement in grammatical English is required in places e.g. line 22-23; Line 67 – replace evidences with evidence
Response 2: We sent the manuscript for English editing and new edited version is uploaded
Point 3: Title – mRCC – use full definition rather than abbreviation
Response 3: We corrected it, thanks
Point 4: Abstract (line 18)– replace abbreviations ICIs and TKI with full definition
Response 4: We corrected it, thanks (line 18-19)
Point 5: Line 26 – typo
Response 5: We corrected “patiens” with patients, thank you (line 28)
Point 6: ‘Pts’ – non-standard abbreviation for patients – replace throughout
Response 6: We have replaced all “pts” with “patients”
Point 7: Replace not responsive patients with non responsive patients
Response 7: Replaced in all manuscript
Point 8: Line 318-19 – unclear, rephrase
Response 8: we have replaced “Among the soluble molecules analyzed, sCTLA-4 resulted the other protein released in serum of mRCC patients associated with a failure to respond to TKi treatment” with “sCTLA4 was the other soluble protein that showed a significant correlation with failure to response during TKi treatment”, Line 350-351
Point 9: For Box & Whisker plots show individual data points
Response 9: We think that those graphs are more clear as box plots
Point 10: 8 patients received sunitinib and 12 patients received pazopanib - the latter is of broader TKI selectivity - the authors should comment on whether any differences in any of the putative biomarkers were obtained between the two different TKIs, and if so why this may be
Response 10: We thank the reviewer for the comment. We did not find any significant difference between the two TKI treated patient groups, although we feel that a larger cohort of patients would be required to address this point due to the different affinity of pazopanib/sunitinib for the receptors among patients.
We have integrated the result section, line 216 as follow: “The results were independent by the TKI administrated (data not shown) and no significant data were obtained for other soluble factors tested (cytokine and checkpoint related protein; Table S2)”

Reviewer 3 Report
The present manuscript by Zizzari et al is a well designed and well executed work. In this work, the authors assessed the immune effect of TKI therapy in order to evaluate the immune status of mRCC patients so that we could define a combination of immunological biomarkers relevant to improve patient’s outcome. They profiled the circulating levels in 20 mRCC patients of exhausted/activated/regulatory T cell subsets through flow cytometry and of 14 immune checkpoint-related proteins and 20 inflammation 27 cytokines/chemokines, using multiplex Luminex assay, at baseline and during TKi therapy. They identified the CD3+CD8+CD137+ and CD3+CD137+PD1+ T cell populations, and 7 soluble immune molecules (IFNγ, sPDL2, sHVEM, sPD1, sGITR, sPDL1 and sCTLA4) associated with clinical response of mRCC patients and/or modulated by TKi therapy.
I would like to say that it is a very well executed work. I have the following queries.
In figure 1C, the authors detected the expression level of PD1+ in CD3+CD137+ T cells, out of curiosity, I would like to ask the authors what was the expression level PD1+ in CD8+CD137+T cells? Have the authors checked this or not?
In line 193-195, the authors quote “Figure 3A shows that before beginning TKi therapy, mRCC patients who will benefit of the treatment had a significantly lower concentration of IFNγ compared to not responsive patients (27.47±8.5 pg/ml for R vs. 515.8±210.6 pg/ml for NR; p=0.007).” How did the authors come to the conclusion that which group will be benefiting in the future? I would like to suggest the authors to represent the results in simpler terms than in so assumptive ones.
As the survival with respect to IFNγ expression and TKi treatment is not correlative then what are the authors emphasizing through Figure 3B?
Minor errors
A period is missing in line 122
Line 151 - …….the levels of on immune checkpoint….. What do the levels of on mean?
Line 26 – Grammatical error
Figure 1C: p=0.04 AND not 0,04; 0.02 and not 0,02
Line 187 – Why is THE in caps lock on mode
Line 204 – IFNγ value is missing
Author Response
Response to Reviewer 3 Comments
Point 1: In figure 1C, the authors detected the expression level of PD1+ in CD3+CD137+ T cells, out of curiosity, I would like to ask the authors what was the expression level PD1+ in CD8+CD137+T cells? Have the authors checked this or not?
Response 1: Thank you to reviewer for indications. We have checked the expression level of PD1+ on CD8+CD137+ T cells and we observed an increase of PD1+ in NR patients both between baseline and during TKi treatment both compared to R group at >T0, but these are trends, no statistically significant data (p=0.08 and p=0.06 respectively). Probably this is for the limited sample size, most probably increasing the number of patients we could obtain an interesting data in the future.
Point 2: In line 193-195, the authors quote “Figure 3A shows that before beginning TKi therapy, mRCC patients who will benefit of the treatment had a significantly lower concentration of IFNγ compared to not responsive patients (27.47±8.5 pg/ml for R vs. 515.8±210.6 pg/ml for NR; p=0.007).” How did the authors come to the conclusion that which group will be benefiting in the future? I would like to suggest the authors to represent the results in simpler terms than in so assumptive ones.
Response 2: Thank you very much for the suggestion. We simply want to say that the patients who will be responsive to TKI treatment have a concentration of IFNγ significantly lower compared to NR group even at baseline.
Point 3: As the survival with respect to IFNγ expression and TKi treatment is not correlative then what are the authors emphasizing through Figure 3B?
Response 3: With figure 3B we want show that low levels of IFNγ at baseline (but not during therapy, >T0) correlate with a better response to TKI treatment in terms of duration of response to therapy (p=0.04).
Minor errors
A period is missing in line 122: Added
Line 151 - …….the levels of on immune checkpoint….. What do the levels of on mean?: Corrected, we eliminated “on”
Line 26 – Grammatical error: Corrected
Figure 1C: p=0.04 AND not 0,04; 0.02 and not 0,02: Corrected and the figure was replaced
Line 187 – Why is THE in caps lock on mode: Replaced with “the”
Line 204 – IFNγ value is missing: added (Line 232)

Round 2
Reviewer 1 Report
The manuscript has been significantly improved in the revised version and the authors addressed all concerns raised by the reviewers